# Evaluation of the BioFire Gastrointestinal Panel to Detect Diarrheal Pathogens in Pediatric Patients

**DOI:** 10.3390/diagnostics12010034

**Published:** 2021-12-24

**Authors:** Sung Jin Jo, Hyun Mi Kang, Jung Ok Kim, Hanwool Cho, Woong Heo, In Young Yoo, Yeon-Joon Park

**Affiliations:** 1Department of Laboratory Medicine, College of Medicine, The Catholic University of Korea, Seoul 06591, Korea; josj0405@gmail.com (S.J.J.); 21801579@cmcnu.or.kr (J.O.K.); swmnm@naver.com (H.C.); windwoongs@naver.com (W.H.); yiy00@naver.com (I.Y.Y.); 2Department of Pediatrics, College of Medicine, The Catholic University of Korea, Seoul 06591, Korea; pedhmk@gmail.com

**Keywords:** BioFire GI Panel, infectious diarrhea, multiplex PCR

## Abstract

Infectious diarrhea is a global pediatric health concern; therefore, rapid and accurate detection of enteropathogens is vital. We evaluated the BioFire^®^ FilmArray^®^ Gastrointestinal (GI) Panel with that of comparator laboratory tests. Stool samples of pediatric patients with diarrhea were prospectively collected and tested. As a comparator method for bacteria, culture, conventional PCR for diarrheagenic *E. coli*, and Allplex GI-Bacteria(I) Assay were tested. For discrepancy analysis, BD MAX Enteric Bacterial Panel was used. As a comparator method for virus, BD MAX Enteric Virus Panel and immunochromatography was used and Allplex GI-Virus Assay was used for discrepancy analysis. The “true positive” was defined as culture-positive and/or positive results from more than two molecular tests. Of the 184 stool samples tested, 93 (50.5%) were true positive for 128 pathogens, and 31 (16.9%) were positive for multiple pathogens. The BioFire GI Panel detected 123 pathogens in 90 of samples. The BioFire GI Panel demonstrated a sensitivity of 100% for 12 targets and a specificity of >95% for 16 targets. The overall positive rate and multiple pathogen rate among patients in the group without underlying diseases were significantly higher than those in the group with hematologic disease (57.0% vs. 28.6% (*p* = 0.001) and 20.4% vs. 4.8% (*p* = 0.02), respectively). The BioFire GI Panel provides comprehensive results within 2 h and may be useful for the rapid identification of enteropathogens.

## 1. Introduction

Globally, infectious diarrhea is associated with childhood morbidity and mortality [1]. As bacteria, viruses, and parasites act on the gastrointestinal tract via various mechanisms, appropriate management of these pathogens requires timely identification of the causative agents [2,3]. Microbiological testing of stool specimens using cultures, enzyme immunoassays (EIA), multiplex molecular assays, and parasite examinations is recommended for the detection of a wide range of causative enteropathogens [4]. However, conventional diagnostic methods such as stool culture can only detect a limited number of pathogens and require long turnaround times, and those including EIAs are not highly sensitive [2,5,6]. Rapid multiplex molecular assays have a rapid turnaround time and high sensitivity [6].

The BioFire^®^ FilmArray^®^ Gastrointestinal (GI) Panel (BF-GI, BioFire Diagnostics, LCC, Salt Lake City, UT, USA) is a random-access assay that simultaneously detects 22 gastrointestinal pathogens on a single platform and requires approximately 2 h to obtain the results. Although there are several studies which evaluated BF-GI, to our knowledge, none of them evaluated all the pathogens with another molecular assay and discrepancy analysis with third party molecular assay in pediatric patients.

In this study, we evaluated BF-GI with comparator molecular assay for all the pathogens included in BF-GI and discrepancy analysis was done with another molecular assay.

## 2. Materials and Methods

### 2.1. Clinical Stool Sample Collection

Pediatric patients with diarrhea, who visited Seoul St. Mary’s Hospital (The Catholic University of Korea, Seoul, Korea) during the study period, were enrolled. Stool samples collected from eligible pediatric patients with diarrhea were submitted to the laboratory between 16 October 2019 and 19 August 2020. Routine laboratory tests and the BF-GI test were ordered for patients with diarrhea that started within 72 h of presentation. Samples were immediately subjected to routine tests (stool culture, immunochromatographic test (ICT), and bacterial molecular tests), as well as the BF-GI test. Only the results of stool samples obtained during the initial diarrhea episode were included in the analysis. Demographic and clinical information such as birthdate, sex, admission date, sample submission time, history of antibiotic treatment, and underlying diseases was retrieved from medical records. The remaining stool samples were stored at −70 °C for viral molecular tests and discrepancy analyses. This study was approved by the Institutional Review Board of Seoul St. Mary’s Hospital (KC18TESI0465). Informed consent was obtained from the parents/guardians of all participants.

### 2.2. BioFire GI Panel Assay

The BF-GI test kit contains all necessary reagent for sample preparation, PCR, and detection. Nucleic acid was extracted and purified from the unprocessed specimens and multiplex PCR processes were performed in the BF-GI system automatically. The system software analyzes the endpoint melting curve for each target on the panel. The BF-GI simultaneously detects 22 pathogens, including 13 bacteria, 5 viruses, and 4 parasites, on a single platform. The bacterial and viral targets of the BF-GI and the comparative tests are described in Table 1. A 200 µL aliquot of stool suspension in Cary-Blair transport medium was used for the BF-GI test, which was conducted according to the manufacturer’s instructions. Specimens were submitted in a stool transport container for stool culture, molecular tests and an immunochromatography test (ICT).

### 2.3. Bacterial Stool Culture

Stool cultures for *Salmonella* spp., *Shigella* spp., *Yersinia* spp., *Campylobacter* spp., *Vibrio* spp., and *Plesiomonas shigelloides* were prepared using an aliquot of stool inoculated on MacConkey agar, Hektoen Enteric agar, Campylobacter CVA agar, thiosulfate-citrate-bile salt agar, cefsulodin-irgasan-novobiocin (CIN) agar, and gram-negative broth, respectively. The inoculated media were incubated in ambient air at 35 °C, except for CIN agar, which was held at room temperature, and campylobacter CVA agar, which was held at 42 °C under microaerobic growth conditions. The plates were held for 3 d, and suspicious colonies were identified using Vitek MS (bioMérieux, Marcy L’Etoile, France) or Vitek 2 (bioMérieux). The presence of *Salmonella* spp. was further confirmed by serotyping with Salmonella Antisera (Joongkyeom, Goyang, Korea).

### 2.4. Molecular Tests for Bacteria

To identify *C. difficile*, *Campylobacter*, *Salmonella*, *Vibrio*, *Y. enterocolitica*, and *Shigella* spp./entero-invasive *Escherichia coli* (EIEC), an Allplex GI-Bacteria(I) Assay (Seegene, Seoul, Korea) was conducted according to the manufacturer’s instructions. Nucleic acid was extracted from stool specimens using a QIAamp DNA Stool Mini Kit (QIAGEN, Hilden, Germany). For *C. difficile*, an Xpert *C. difficile* assay kit (Cepheid, Sunnyvale, CA, USA) was also used. Conventional PCR (diarrheagenic *E. coli* PCR) was used to detect diarrheagenic *E. coli*. DNA was extracted using the QIAamp DNA Mini Kit (QIAGEN) according to the manufacturer’s instructions. PCR primers were used as previously described [7]. This PCR detects *sep*A and *agg*R in enteroaggregative *E. coli* (EAEC), *esc*V and *eae*A in enteropathogenic *E. coli* (EPEC), *estib* and *elt* in enterotoxigenic *E. coli* (ETEC), *stx1*/*stx2* in Shiga toxin-producing *E. coli* (STEC), and *ipa*H in EIEC.

### 2.5. Immunochromatography Test for Rotavirus and Norovirus

CareUS Rotavirus Plus (Wells Bio, Seoul, Korea) and CerTest Norovirus (Biotec S.L., Zaragoza, Spain) tests were performed according to the manufacturer’s instructions to detect rotavirus and norovirus viral antigens, respectively.

### 2.6. Molecular Tests for Viruses

BD MAX Enteric Viral Panel (BD Diagnostics, Baltimore, MD, USA) was used to identify rotavirus, norovirus, adenovirus, astrovirus and sapovirus. A 5 µL sample was placed into the sample buffer and was loaded into the BD MAX instrument. Sample preparation, lysis, extraction of the nucleic acid and multiplex PCR were performed automatically with the BD MAX system.

### 2.7. Discrepancy Analysis

“True positive” was defined as a positive result from at least two molecular tests, including comparator tests and the BF-GI. Culture-positive was considered a positive result, regardless of other results. “True negative” was defined as a negative result from at least two molecular assays in which the organisms concerned were included as a target of the assays.

Any discrepancies between the BF-GI and comparator molecular tests for bacteria were analyzed using the BD MAX Enteric Bacterial Panel (BD Diagnostics) (Table 1). For EPEC and EAEC, which are not included in the BD MAX Enteric Bacterial Panel, an additional in-house PCR test [8] was performed. For a discrepancy between the BF-GI and BD MAX Enteric Viral Panel, the Allplex GI-Virus Assay (Seegene Inc., Seoul, Korea) was used.

### 2.8. Statistical Analysis

All statistical analyses were performed using MedCalc version 20 (MedCalc Software Ltd., Ostend, Belgium). The Kolmogorov–Smirnov test was used to determine the normal distribution of enrolled patients. Cohen’s kappa coefficient (*k*) was used to measure inter-rate reliability for agreement among assays, where a kappa value of <0 indicated no agreement and a kappa value between 0 and 0.20 as slight, 0.21 and 0.40 as fair, 0.41 and 0.60 as moderate, 0.61 and 0.80 as substantial, and 0.81 and1 as almost perfect agreement. Sensitivity and specificity were calculated using the χ^2^ test or Fisher’s exact test. Statistical significance was set at *p* < 0.05.

## 3. Results

### 3.1. Patients Characteristics

The characteristics of patients (*N* = 184) are described in Table 2. The average age of the study population was 4.0 (median ± 4.7; interquartile range 1–8.5) y, and all were inpatients. Among these patients, 142 had no underlying disease and 42 had hematologic diseases (eight received HSCT).

### 3.2. Results of the BioFire GI Panel Assay

Overall, 128 pathogens were detected in 93 (50.5%) samples and were determined as true positive. The most frequently detected pathogen was norovirus (19.0%), followed by *C. difficile* (12.0%), *Campylobacter* spp. (10.9%), EPEC (7.6%), rotavirus (6.0%), *Salmonella* spp. (3.3%), *Y. enterocolitica* (2.7%), STEC (1.6%), sapovirus (1.6%), astrovirus (1.1%), ETEC (0.5%) and *P. shigelloides* (0.5%) (Table 3). No parasites were detected.

The BF-GI detected 123 pathogens in 90 (48.9%) samples and demonstrated a 92.4% (167/184) overall concordant rate with true positive and true negative results. Compared with the comparator tests, the BF-GI detected three additional bacteria (1 EPEC, 1 STEC, and 1 *P. shigelloides*). Among the 90 pathogen-positive samples, one pathogen was detected in each of 61 samples (33.2%), two in each of 25 samples (13.6%), and three in each of 4 samples (2.2%). The sensitivity, specificity, and *k* values obtained from the BF-GI for each detectable pathogen are shown in Table 3. These indicated 100% sensitivity for 12 pathogens and 100% specificity for 10 pathogens, and where the specificity for all pathogens was >97%. There were 14 false-positive results (5 cases of EPEC; 3 of *C. difficile*; two each of *Salmonella* spp. and EAEC; and one each of *Shigella* spp./EIEC, and adenovirus) and one false-negative result for rotavirus.

### 3.3. Results of the Comparator Tests

The comparator laboratory tests detected 125 pathogens in 93 samples (50.5%), demonstrating an overall concordance of 95.7% (176/184) with true positive and true negative results. In the 93 pathogen-positive samples, 1 pathogen was detected in each of 64 samples (34.7%), 2 pathogens were detected in each of 26 samples (14.1%), and 3 pathogens were detected in each of 3 samples (1.6%) (Table 2).

Stool culture indicated a sensitivity of 85.0% for *Campylobacter* spp., 100% for *Salmonella* spp., 60.0% for *Y. enterocolitica*, and 0% for *P. shigelloides*. Stool culture indicated a specificity of 100% for five targeted pathogens (Table 3). There were six false-negative results—3 for *Campylobacter* spp., 2 for *Y. enterocolitica*, and 1 for *P. shigelloides*. Stool culture detected two additional isolates of *Y. pseudotuberculosis*, which was not included in the BF-GI target.

The Allplex GI-Bacteria(I) Assay yielded a *k* value of >0.9 and sensitivity of >90% for detecting *C. difficile*, *Campylobacter* spp., *Salmonella* spp., and *Y. enterocolitica* (Table 3). Although specificity was >99% for all pathogens, there was 1 false-positive case of *Campylobacter* spp., and 1 false-negative of *C. difficile*. Two cases of *Aeromonas* spp. (not a comparative target) were detected via the Allplex GI-Bacteria (I) Assay. Xpert *C. difficile* showed a *k* value of 0.947, sensitivity of 90.9%, and specificity of 100%, with two false-negative cases. Conventional PCR for diarrheagenic *E. coli* showed sensitivity of 92.9% for EPEC, 100% for ETEC, and 66.7% for STEC. Specificity was >98% for five diarrheagenic *E. coli*. There were 2 false-positive cases of EAEC, and 1 false-negative case each of EPEC and STEC (Table 4).

The BD MAX Enteric Viral Panel showed sensitivity of 100% for enteric viruses, and specificity was >99%. There was 1 false-positive case each of adenovirus and astrovirus. CareUS Rotavirus Plus showed a *k* value of 0.243, sensitivity of 18.2%, and specificity of 98.8% with 2 false-positive and 9 false-negative cases. CerTest Norovirus showed a *k* value of 0.577, sensitivity of 45.7%, and specificity of 100%, with 19 false-negative cases.

### 3.4. Co-Detected Multiple Pathogens in a Sample

Using the BF-GI and comparator methods, multiple pathogens were detected in 31 samples (Table 5). Norovirus was the most frequently detected multiple pathogen (16/184, 8.7%), followed by *C. difficile* (14/184, 7.6%) and EPEC (10/184, 5.4%). *Campylobacter* spp., the most commonly detected bacterium, showed a relatively low co-detection rate (9/184, 4.9%). The distribution of multiple pathogen samples varied according to age group. Multiple pathogens were significantly higher in the 0–5 y group (22.9%) than in the ≥6 y group (8.0%; *p* = 0.008; 95% CI = 4.1–24.6). The multiple pathogen sample rates of patients without underlying diseases were significantly higher than those of patients with hematology diseases (20.4% vs. 4.8%; *p* = 0.004; 95% CI = 7.1–25.9).

### 3.5. Distribution of Pathogens According to Age Groups and Medical Conditions

The distribution of gastrointestinal pathogens according to age groups and medical conditions is shown in Table 6. The positive rate was high in the 1–5 y group, although there were no significant differences between groups. The positive rate of *C. difficile* gradually decreased with age. The positive rate of *Campylobacter* spp. was significantly higher in patients aged over 5 y (18.6%; 14/75) than ≤5 y (5.5%; 6/109) (*p* = 0.005; 95% CI = 3.7–23.8), whereas rotavirus and norovirus were the major pathogens in patients aged ≤ 5 y. The detection rates for rotavirus (9.2% vs. 1.3%) and norovirus (28.4% vs. 5.3%) were significantly higher in patients aged ≤ 5 y (*p* = 0.03, 95% CI = 0.7–14.9, *p* = 0.001, 95% CI = 12.5–32.7, respectively) than in those aged > 5 y.

According to the presence or absence of a hematology disease, 113 pathogens (68 bacteria; 47 viruses) were detected in 142 patients without underlying diseases, and 15 pathogens (10 bacteria; 5 viruses) were detected in 42 patients with hematology disease. The overall positive rate of the former (57.0%; 81/142) was significantly higher than that in the latter (28.5%; 12/42); (*p* = 0.001; 95% CI = 11.4–42.3). The differences in the positive rates between groups with and without hematology disease were significant for bacteria (38.0%; 54/142 vs. 16.6%; 7/42; *p* = 0.01; 95% CI = 5.6–33.1) and for viruses (28.9%; 41/142 vs. 11.9%; 5/42; *p* = 0.03; 95% CI = 2.5–27.4).

## 4. Discussion

Overall, BF-GI showed high sensitivity and specificity in detecting diarrheal pathogens. Compared with the comparator methods, the BF-GI additionally detected 3 bacteria. Comparator methods detected four bacterial isolates which were not included in BF-GI targets.

In the detection of bacterial pathogens, both BF-GI and Allplex GI-Bacteria(I) Assay showed overall high sensitivities. However, BF-GI showed a few false-positive results for *Salmonella* spp. Although molecular assays were sensitive to *Salmonella*, which was equivalent to or higher than that of the stool culture [9,10], many false-positives and/or false-negatives were observed even between molecular methods when detecting *Salmonella* spp. The reason for this may be asymptomatic carriage of *Salmonella* spp. or cross reactivity with an alternate component of the gastrointestinal microbiota [11,12,13].

In addition, the BF-GI showed a few false-positive results for diarrheagenic *E. coli*, mainly with EPEC. The most frequently observed error in this study was the five false-positive cases of EPEC. Although EPEC infection is reportedly less severe and is more sporadic than other diarrheal illnesses [14], as EPEC outbreaks caused by contaminated food have been reported in the Republic of Korea [15], detection of diarrheagenic *E. coli* has attained greater importance recently. Nevertheless, as there are few commercial diarrheagenic *E. coli* PCR kits available, we conducted another conventional PCR test [8] to confirm EAEC and EPEC, the accuracy of which may have been relatively inadequate.

*Campylobacter* spp. was the most frequently detected pathogen in patients aged > 5 y. The rate of *Campylobacter* spp. isolation in pediatric patients has increased compared with that of *E. coli* or *Salmonella* spp., in our country, and the introduction of multiple molecular assays may be attributed to the increasing rate of *Campylobacter* spp. isolation [16,17]. In the present study, the rate of detection of *Campylobacter* spp. varied with age groups: <1 y (0%), 1–5 y (8.9%), 5–10 y (20.0%), and ≥ 11 y (17.1%); the higher the age, the higher the prevalence of *Campylobacter* spp. These results concurred with previous findings and may be attributed to changes in eating habits with increasing age [17]. In the present study, among the 20 cases of *Campylobacter* spp. detected by molecular assays, 17 were also detected by culture test where all 17 isolates were identified as *C. jejuni*. The higher sensitivity of the BioFire panel for *Campylobacter* spp. than that of culturing has been reported [18]. In the present study, relatively high prevalence of *Campylobacter* spp. infection observed in patients with hematologic disease (7.1%), and considering the fastidious characteristics of *Campylobacter* spp. to be cultured [19], the molecular methods will be very useful.

For virus detection, the molecular tests showed better performance than ICTs. It is known that ICTs, which generally show sensitivities between ~30% and 90%, are less sensitive than molecular assays [20,21]. The results of the BF-GI and the BD MAX Viral Panel showed almost perfect agreement and also showed good discrimination capabilities of multiple viral pathogens. Since ICT tests have low sensitivity for rotavirus/norovirus and there were considerable detection rates for adenovirus, astrovirus, and sapovirus in this study, multiplex PCR is expected to be useful in enteric viral detection.

Regarding medical status, positivity rates were higher in patients without underlying diseases (57.0%) than in those with hematologic diseases (30.3%). This result is consistent with that of Stockman et al. [22], who reported that children with underlying diseases (cardiovascular, malignancy, gastrointestinal, and respiratory diseases) were less likely to have gastrointestinal pathogens than those without underlying diseases (45% vs. 60%, respectively). In the present study, 42 had hematologic diseases, of which eight underwent HSCT. Of the eight patients who underwent HSCT, pathogens were detected in only one sample (1 case of norovirus). The cause of diarrhea in this group was different from that in typical diarrhea patients. Graft-versus-host disease, viral enteritis, and *C. difficile* were the main causes of diarrhea in pediatric patients who underwent HSCT. HSCT as well as chemotherapy may cause changes in the intestinal microbiome diversity [23,24]. However, though the positive rate of diarrhea pathogen was low in hematology disease group, critical pathogens such as *Campylobacter* spp., *Salmonella* spp., and STEC were also detected considerably. In addition, it was surveyed that the very low proportion (25%) of clinical microbiology laboratories in Korea cultivate *Campylobacte*r [25].

This study has a limitation in that it was performed in a single center. Seoul St. Mary’s Hospital is a tertiary university hospital providing treatment for high-risk pediatric patients through its pediatric Hemato-Oncology Center, and this may have resulted in lower pathogen-positive rates. Nevertheless, the current study showed the characteristics of diarrhea-causing pathogens in high-risk pediatric patients compared to those without underlying diseases.

In conclusion, our results indicate that the BF-GI demonstrated 100% sensitivity for 12 pathogens, and >97% specificity for 16 pathogens. Considering the short time required to obtain results and broad pathogen targets using the BioFire panel, it may be useful for the rapid identification of GI pathogens.

## Figures and Tables

**Table 1 diagnostics-12-00034-t001:** Bacterial and viral pathogens targeted by the BioFire GI Panel, comparator tests, and discrepancy analysis tests.

Targets	Comparator Tests		Discrepancy Analysis
*C. difficile*	Xpert *C. difficile*	Allplex Bacteria	
*Campylobacter* spp.	Stool culture	Allplex Bacteria	BD Bacterial
*Salmonella* spp.	Stool culture	Allplex Bacteria	BD Bacterial
*Vibrio* spp.	Stool culture	Allplex Bacteria	BD Bacterial
*Y. enterocolitica*	Stool culture	Allplex Bacteria	BD Bacterial
*Plesiomonas shigelloides*	Stool culture		BD Bacterial
EAEC		*E. coli* PCR	secondary *E. coli* PCR
EPEC		*E. coli* PCR	secondary *E. coli* PCR
ETEC		*E. coli* PCR	BD Bacterial
STEC		*E. coli* PCR	BD Bacterial
*Shigella* spp./EIEC		Allplex Bacteria/*E. coli* PCR	BD Bacterial
*E. coli* O157			secondary *E. coli* PCR
Rotavirus	careUS Rotavirus Plus	BD Viral	Allplex Virus
Norovirus GI/GII	CerTest Norovirus	BD Viral	Allplex Virus
Adenovirus F40/41		BD Viral	Allplex Virus
Astrovirus		BD Viral	Allplex Virus
Sapovirus		BD Viral	Allplex Virus

Abbreviations: Allplex Bacteria, Seegene Allplex GI-Bacteria(I) Assay; BD Bacterial, BD MAX Enteric Bacterial Panel; BD Viral, BD MAX Enteric Viral Panel; Allplex Virus, Seegene Allplex GI-Virus Assay; EAEC, enteroaggregative *Escherichia coli*; EPEC, enteropathogenic *E. coli*; ETEC, enterotoxigenic *E. coli*; STEC, Shiga toxin-producing *E. coli*; EIEC, enteroinvasive *E. coli*.

**Table 2 diagnostics-12-00034-t002:** Characteristics of 184 patients and summary of the detected pathogens.

Characteristics				No.	Consensus Results		
					No. of Positive Samples	No. of Detected Pathogens (Sample No.: 1 Isolate/2 Isolates/3 Isolates)	No. of Bacteria/Virus
Age	<1 y			42	18/42 (42.9%)	27 (11/5/2)	11/16
	1–5 y			67	41/67 (61.2%)	60 (23/17/1)	30/30
	6–10 y			40	20/40 (50.0%)	24 (17/2/1)	20/4
	≥11 y			35	14/35 (40.0%)	17 (11/3/0)	15/2
Medical condition	Without-underlying disease			142	81/142 (57.0%)	113 (52/26/3)	68/47
	Hematology			42	12/42 (28.6%)	15 (10/1/1)	10/5
			HSCT-	34	11/34 (32.4%)	14 (9/1/1)	10/4
			HSCT+	8	1/8 (12.5%)	1 (1/0/0)	0/1
BF-GI					90/184 (48.9%)	123 (61/25/4)	72/51
C					93/184 (50.5%)	125 (64/26/3)	73/52
Total				184	93/184 (50.5%)	128 (62/27/4)	76/52

Abbreviations: BF-GI, BioFire GI Panel; C, comparator tests; HSCT, hematopoietic stem cell transplantation.

**Table 3 diagnostics-12-00034-t003:** Results of the BioFire GI Panel and comparator laboratory tests.

Pathogen	No.	BioFire GI Panel				Comparator Tests			
		κ	Sensitivity (95% CI)	Specificity (95% CI)	TP/FP/TN/FN	κ	Sensitivity (95% CI)	Specificity (95% CI)	TP/FP/TN/FN
*C. difficile*		0.927	100 (84.6–100)	98.2 (94.7–99.6)	22/3/159/0	0.946 ^a^	90.9 (70.8–98.9)	100 (97.9–100)	20/0/162/2
						0.974 ^b^	95.5 (77.2–99.9)	100 (97.8–100)	21/0/162/1
*Campylobacter* spp.	20	1.000	100 (83.2–100)	100 (97.8–100)	20/0/164/0	0.910 ^c^	85.0 (62.1–96.8)	100 (97.8–100)	17/0/164/3
						0.973 ^b^	100 (83.2–100)	99.4 (96.7–99.9)	20/1/163/0
*Salmonella* spp.	6	0.852	100 (54.1–100)	98.9 (96.0–99.9)	6/2/176/0	1.000 ^c^	100 (54.1–100)	100 (98.0–100)	6/0/178/0
						1.000 ^b^	100 (54.1–100)	100 (98.0–100)	6/0/178/0
*Vibrio* spp.	0	NA	NA	100 (98.0–100)	0/0/184/0	NA ^c^	NA	100 (98.0–100)	0/0/184/0
						NA ^b^	NA	100 (98.0–100)	0/0/184/0
*Y. enterocolitica*	5	1.000	100 (47.8–100)	100 (98.0–100)	5/0/179/0	0.745 ^c^	60.0 (14.7–94.7)	100 (98.1–100)	3/0/179/2
						1.000 ^b^	100 (47.8–100)	100 (98.0–100)	5/0/179/0
*P. shigelloides*	1	1.000	100 (2.5–100)	100 (98.0–100)	1/0/183/0	0.000 ^c^	0 (0–97.5)	100 (98.0–100)	0/0/183/1
EAEC	0	0.000	NA	98.9 (96.1–99.9)	0/2/182/0	0.000 ^d^	NA	98.9 (96.1–99.9)	0/2/182/0
EPEC	14	0.843	100 (78.2–100)	97.0 (93.2–99.0)	15/5/164/0	0.960 ^d^	92.9 (66.1–99.8)	100 (97.9–100)	13/0/170/1
ETEC	1	1.000	100 (2.5–100)	100 (98.0–100)	1/0/183/0	1.000 ^d^	100 (2.5–100)	100 (98.0–100)	1/0/183/0
STEC	3	1.000	100 (29.2–100)	100 (98.0–100)	3/0/181/0	0.797 ^d^	66.7 (9.4–99.2)	100 (98.0–100)	2/0/181/1
*Shigella* spp./EIEC	0	0.000	NA	99.5 (97.0–99.9)	0/1/183/0	NA ^b,d^	NA	100 (98.0–100)	0/0/184/0
Rotavirus	11	1.000	100 (71.5–100)	100 (97.9–100)	11/0/173/0	0.243 ^e^	18.2 (2.3–51.8)	98.8 (95.9–99.8)	2/2/171/9
						1.000 ^f^	100 (71.5–100)	100 (97.9–100)	11/0/173/0
Norovirus GI/GII	35	0.982	97.1(85.1–99.9)	100 (97.6–100)	34/0/149/1	0.577 ^e^	45.7 (28.8–63.4)	100 (97.6–100)	16/0/149/19
						1.000 ^f^	100 (90.5–100)	100 (97.6–100)	35/0/149/0
Adenovirus F40/41	1	0.664	100 (2.5–100)	99.5 (97.0–99.9)	1/1/182/0	0.664 ^f^	100 (2.5–100)	99.5 (97.0–99.9)	1/1/182/0
Astrovirus	2	1.000	100 (15.8–100)	100 (98.0–100)	2/0/182/0	0.797 ^f^	100 (15.8–100)	99.5 (97.0–99.9)	2/1/181/0
Sapovirus	3	1.000	100 (29.2–100)	100 (98.0–100)	3/0/181/0	1.000 ^f^	100 (29.2–100)	100 (98.0–100)	3/0/181/0

^a^ Results of Xpert *C. difficile* assay, ^b^ Result of Allplex GI-Bacteria(I), ^c^ Result of culture, ^d^ Result of *E. coli* PCR, ^e^ Result of immunochromatographic test, ^f^ Result of BD MAX Enteric viral Panel. Abbreviations: κ, Cohen’s kappa coefficient; TP, true-positive; FP, false-positive; TN, true-negative; FN, false-negative; NA, not available; CI, confidence interval.

**Table 4 diagnostics-12-00034-t004:** Discrepancy analysis of bacteria detected via the FilmArray GI Panel and comparator tests.

Sample No.	FilmArray GI Panel	Comparator Tests			Discrepancy Analysis	
		Culture	Allplex GI-Bacteria(I)	Conventional *E. coli* PCR	BD MAX Enteric Bacterial Panel	Secondary *E. coli* PCR
1	Negative	Negative	*Salmonella* spp. *	Negative	Negative	
2	*Campylobacter* spp./EPEC *	*C. jejuni*	*Campylobacter* spp.	Negative		Negative
11	Negative	Negative	*Campylobacter* spp. *	Negative	Negative	
12	EPEC *	Negative	Negative	Negative		Negative
18	*C. difficile/*EAEC *	*C. difficile*	*C. difficile*	Negative		Negative
50	*Campylobacter* spp.	Negative	*Campylobacter* spp.	EAEC *		Negative
76	EPEC *	Negative	Negative	Negative		Negative
91	EPEC *	Negative	Negative	Negative		Negative
113	EAEC *	Negative	Negative	Negative		Negative
131	EPEC	Negative	Negative	EPEC/EAEC *		Negative
141	*Salmonella* spp./*P. shigelloides*	*Salmonella* group B	*Salmonella* spp.	Negative	*Salmonella* spp./*P. shigelloides*	
144	STEC/EPEC	Negative	Negative	STEC/EPEC		
154	*Salmonella* spp. *	Negative	Negative	Negative	Negative	
158	EPEC *	Negative	Negative	Negative		Negative
160	*Campylobacter* spp./EPEC	*C. jejuni*	*Campylobacter* spp.	Negative ^†^		EPEC
177	*C. difficile/*STEC/EPEC	*C. difficile*	*C. difficile*	EPEC/Negative ^†^	STEC	
182	*Salmonella* spp.*	Negative	Negative	Negative	Negative	
183	*Salmonella* spp./EIEC *	*Salmonella* group C	*Salmonella* spp.	*Salmonella* spp.	*Salmonella* spp.	

* false positive, ^†^ false negative.

**Table 5 diagnostics-12-00034-t005:** Distribution of co-detected pathogens according to age group and medical condition.

Pathogens			No. of Samples According to Age Groups				Medical Condition		Total
			<1 y	1–5 y	6–10 y	≥11 y	Without Underlying	Hematology Disease	
Norovirus			4	12	3	0	16	3	19
Norovirus	*C. difficile*		3	6	0	1	10	0	10
Norovirus	*Campylobacter* spp.		0	1	0	0	1	0	1
Norovirus	Rotavirus		0	1	0	0	1	0	1
Norovirus	EPEC		0	1	0	0	1	0	1
Norovirus	Sapovirus		0	1	0	0	1	0	1
Norovirus	Rotavirus	Sapovirus	1	0	0	0	1	0	1
Norovirus	Sapovirus	Astrovirus	0	1	0	0	1	0	1
*C. difficile*			2		4	2	5	3	8
*C. difficile*	Norovirus		3	6	0	1	10	0	10
*C. difficile*	Rotavirus		2	0	0	0	2	0	2
*C. difficile*	*Campylobacter* spp.		0	1	0	0	1	0	1
*C. difficile*	STEC	EPEC	1	0	0	0	0	1	1
EPEC			1	2	1	0	4	0	4
EPEC	*Campylobacter* spp.		0	1	2	0	3	0	3
EPEC	STEC		0	2	0	0	2	0	2
EPEC	*Y. enterocolitica*		0	1	0	1	2	0	2
EPEC	Norovirus	Norovirus	0	1	0	0	1	0	1
EPEC	*Campylobacter* spp.	ETEC	0	0	1	0	1	0	1
EPEC	STEC	*C. difficile*	1	0	0	0	0	1	1
*Campylobacter* spp.				2	5	5	10	2	12
*Campylobacter* spp.	EPEC		0	1	2	0	3	0	3
*Campylobacter* spp.	*Aeromonas* spp.		0	1	0	1	2	0	2
*Campylobacter* spp.	EPEC	ETEC	0	0	1	0	1	0	1
*Campylobacter* spp.	*C. difficile*		0	1	0	0	1	0	1
*Campylobacter* spp.	Norovirus		0	1	0	0	1	0	1
*P. shigelloides*	*Salmonella* spp.		0	1	0	0	0	1	1
Co-detection rate (No. of co-detected samples/total samples)			16.7% (7/42)	26.9% (18/67)	7.5% (3/40)	8.6% (3/35)	20.4% (29/142)	4.8% (2/42)	16.9% (31/184)

Abbreviations: EPEC, enteropathogenic *Escherichia coli*; ETEC, enterotoxigenic *E. coli*; STEC, Shiga toxin-producing *E. coli*.

**Table 6 diagnostics-12-00034-t006:** Number (%) of pathogens according to age group and medical condition.

Pathogen	Age Group				Medical Condition		Total
	<1 y	1–5 y	6–10 y	≥11 y	Without Underlying	Hematology Disease	
*C. difficile*	8 (19.1%)	7 (10.5%)	4 (10.0%)	3 (8.6%)	18 (12.7%)	4 (9.5%)	22
*Campylobacter* spp.	-	6 (8.9%)	8 (20.0%)	6 (17.1%)	18 (12.7%)	3 (7.1%)	20
*Salmonella* spp.	-	4 (6.0%)	2 (5.0%)	-	5 (3.5%)	1 (2.4%)	6
*Y. enterocolitica*	-	1 (1.5%)	1 (2.5%)	3 (8.6%)	5 (3.5%)	-	5
*P. shigelloides*	-	1 (1.5%)	-	-	-	1 (2.4%)	1
EPEC	2 (4.8%)	7 (10.5%)	4 (10.0%)	1 (2.9%)	13 (9.2%)	2 (4.7%)	14
ETEC	-	-	1 (2.5%)	-	1 (0.7%)	-	1
STEC	1 (2.4%)	2 (3.0%)	-	-	2 (1.4%)	1 (2.4%)	3
Rotavirus	6 (14.3%)	4 (6.0%)	-	1 (2.9%)	10 (7.0%)	2 (4.7%)	11
Norovirus GI/GII	8 (19.0%)	23 (34.3%)	3 (7.5%)	1 (2.9%)	32 (22.5%)	5 (11.9%)	35
Adenovirus F40/41	1 (2.4%)	-	-	-	1 (0.7%)	-	1
Astrovirus	-	1 (1.5%)	1 (2.5%)	-	1 (0.7%)	1 (2.4%)	2
Sapovirus	1 (2.4%)	2 (3.0%)	-	-	3 (2.1%)	-	3
*Y. pseudotuberculosis* *	-	1 (1.5%)	-	1 (2.9%)	2 (1.4%)	-	2
*Aeromonas* spp. *	-	2 (3.0%)	-	1 (2.9%)	2 (1.4%)	1 (2.4%)	3
No. of detected pathogens (%)	27	61	24	17	113	15	128
No. of positive samples/tested samples (%)	18/42 (42.9%)	42/67 (62.7%)	20/40 (50.0%)	14/35 (40.0%)	81/142 (57.0%)	12/42 (28.6%)	93/184 (50.5%)

Abbreviations: EPEC, enteropathogenic *Escherichia coli*; ETEC, enterotoxigenic *E. coli*; STEC, Shiga toxin-producing *E. coli*. * Pathogens are not included as BioFire GI Panel targets.

## Data Availability

The datasets used and analyzed in the current study are available from the corresponding author upon reasonable request.

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
