# Peer review of "Evaluation of the BioFire Gastrointestinal Panel to Detect Diarrheal Pathogens in Pediatric Patients"

_diagnostics, 2021, doi:10.3390/diagnostics12010034_

Round 1
Reviewer 1 Report
The authors investigated the performance level of “BioFire GI Panel assay” in terms of detection of pathological bacteria and viruses comparing with stool culture, molecule test and immunochromatography test in pediatric patients. They found that BF-GI demonstrated good PPA and NPA for most of targeted pathogens. Although this study includes substantial number of samples, the performance of BF-GI was not analyzed properly due to absence of gold standard for detection of pathogens. Besides, the two sections in the results, “Co-detected multiple pathogens in a sample” and “Distribution of pathogens according to age groups and medical conditions”, should be analyzed based on the results of golden standard.
Major comments:
BF-GI was compared with routine tests including culture, molecular test and ICT regarding their PPA and NPA. However, it is vaguely described about which test was regarded as a golden standard. Please clarify the golden standard and calculate the performance level of BF-GI including sensitivity and specificity for all bacteria and viruses detected in the study. For rotavirus and norovirus, BD max Enteric viral Panel should be performed for all samples if it was the golden standard for detecting these viruses.
Author Response
We thank you for your consideration on our submission (diagnostics-1430886). I appreciate the time and detail provided by each reviewer. The manuscript has certainly benefited from these insightful suggestions.
Fisrt of all, authors clarified the gold standard pointed out by reviewer. The process and description of the changes are as follows.
- To clarify gold standard, we conducted more than three molecular tests, including discrepancy analysis, for all bacteria and viruses. Based on this definition, we calculated sensitivity, specificity based on the true-positive or true-negative results and the term ‘routine test’ was changed to ‘comparator test’ instead.
- We conducted the BD-MAX Virus panel assay for all samples. Through this process, two samples that could not be additionally tested due to insufficient remaining amount were excluded from the study, and the total number of study samples reached 198.
- In addition, the ‘co-detection of multiple pathogens’ and ‘distribution of pathogens according to the age groups and medical conditions’ were analyzed based on the gold standard results.
Reviewer 2 Report
The paper is not publishable in the current form. I carefully read the abstract and expected to find convincing evidence for their findings, the context for the study and implications of the research.
The paper appears to be random discussion of all of their results.
The authors must define a patient group that they study, and then compare with routine culture methods and other molecular methods with the study method. I am not quite certain how to start this process, but the authors must define a hypothesis in the introduction and then assess the outcomes of their testing. It would appear they are documenting literally hundreds of tests that are performed hundreds of patients, and then anecdotally defining interesting observations.
I think the authors have an interesting data base that offers considerable knowledge regarding the applications of these tests, but there is virtually no organization or convincing evidence from all of this data. That is disappointing. One could probably design three or four studies from this data base.
The tables which are my first look, sometimes before I have read the text are unintelligible.
At the end of the day, I must send my colleagues back to the drawing board, and wish them luck. I liked the idea.
Author Response
We thank you for your consideration on our submission (diagnostics-1430886). I appreciate the time and detail provided by each reviewer. The manuscript has certainly benefited from these insightful suggestions.
As pointed out in Reviewer 2, we presented a predictable hypothesis to the patient group who participated in this study and presented it to Introduction.
- In order to present the hypothesis more clearly in our study, the patient group was set up in two groups without underlying disease and with-hematology disease, and 14 other disease groups were excluded from the study.
- Through this process, the number of samples for the study finally reached 184.
- The tables were revised to increase readability.
Reviewer 3 Report
The Authors wrote an interesting manuscript on the evaluation of the BioFire Gastrointestinal Panel to detect diarrhea pathogens in pediatric patients. The manuscript is well written and presents interesting data. Few comments to improve its quality:
- the name of the species should be written in italics.
- Parasites were not detected and no routine screening comparison was reported. For this reason the evaluation should be mentioned as focused (in both text and title) to viral and bacterial diarrhea pathogens.
- The study is based on results of a single center. This limitation should be mentioned in the Discussion section.
Author Response
We thank you for your consideration on our submission (diagnostics-1430886). I appreciate the time and detail provided by each reviewer. The manuscript has certainly benefited from these insightful suggestions.
According to the comment of reviewer 3, italics were applied to the name of species and the limitations of this study were described in the discussion.
Parasites were not found in this study, so the subjects of the study were limited to bacteria and viruses, and it was revised and described in the title and text.
Round 2
Reviewer 1 Report
The authors amended the manuscript enough according to my comments.
Reviewer 2 Report
This paper is much improved; table 4 is not very understandable, and probably could be deleted if its unclear after formatting. The paper though understandable is very complex presenting a very granular about of study detail. I think the important points of the article is the high efficiency of recovering and identifying gut bacteria, and the moderate difference in patients with and without chronic diseases. I like the fact that they include comparator tests to sort out the outliers,.